# SHIFT-TOLERANT PERCEPTUAL SIMILARITY METRIC

## ABSTRACT

Existing perceptual similarity metrics assume an image and its reference are well aligned. As a result, these metrics are often sensitive to a small alignment error that is imperceptible to the human eyes. This paper studies the effect of small misalignment, specifically a small shift between the input and reference image, on existing metrics, and accordingly develops a shift-tolerant similarity metric. This paper builds upon LPIPS, a widely used learned perceptual similarity metric and explores architectural design considerations to make it robust against imperceptible misalignment. Specifically, we study a wide spectrum of neural network elements, such as anti-aliasing filtering, pooling, striding, padding, and skip connection, and discuss their roles in making a robust metric. Based on our studies, we develop a new deep neural network-based perceptual similarity metric[1]. Our experiments show that our metric is tolerant to imperceptible shifts while being consistent with the human similarity judgment.

## 1 INTRODUCTION

Image similarity measurement is a common task for many computer vision and computer graphics applications. General similarity metrics like PSNR and RMSE, however, do not match the human visual perception well when assessing the similarity between two images. Therefore, many dedicated image similarity metrics, such as Structural Similarity (SSIM) and its variations (Wang et al., 2004; 2003; Zhang et al., 2011; Wang & Simoncelli, 2005), were developed in order to more closely reflect the human perception. However, manually crafting a perceptual similarity metric remains a challenging task as it involves the complex human cognitive judgement (Medin et al., 1993; Tversky, 1977; Wang et al., 2004; Zhang et al., 2018).

Recently, learning-based image similarity metrics have been developed. These metrics learn from a large set of labelled data and predict the similarity between images that correlates well with human perception (Bhardwaj et al., 2020; Ding et al., 2020; Kettunen et al., 2019; Prashnani et al., 2018; Zhang et al., 2018; Czolbe et al., 2020). Among them, the Learned Perceptual Image Patch Similarity metric (LPIPS) by Zhang et al. (2018) is now widely adopted as a perceptual similarity metric and used in computer graphics and vision literature.

Figure 1: Whether two images $I_0$ and $I_1$, are shifted by 1-pixel or not, viewers always consider $I_1$ as more similar to $I_{ref}$ than $I_0$. However, existing metrics often switch their predictions after the imperceptible 1-pixel shift.

This paper studies how image similarity metrics work on a pair of images that are not perfectly aligned. For instance, a tiny misalignment in the image pair such as a one-pixel translation between them, is imperceptible to the human eyes. But, *will such a visually imperceptible misalignment compromise any existing similarity metrics*? For PSNR and RMSE, since they assume pixel-wise registration, naturally they are sensitive to as small as a one-pixel misalignment. As we will detail in this paper, our study found that the learned perceptual similarity metrics, such as LPIPS, are also sensitive to a small misalignment. Figure 1 shows such an example through a two-alternative forced choice test. In this test, viewers were asked "*which of the two distorted images, $I_0$ or $I_1$, is more similar to the reference image $I_{ref}$?*" Then, we shifted $I_0$ and $I_1$ by one pixel and obtained their opinions again. None of

---

[1]We will make our code and data publicly available.

the participants flipped their opinions from $I_0$ to $I_1$ or vice versa, which is intuitive as a one-pixel shift is imperceptible to viewers. But existing metrics, such as MS-SSIM and LPIPS, flipped their judgments after the one-pixel shift.

Our problem is related to the recent work on making deep neural networks shift invariant (Islam et al., 2020; Kayhan & Gemert, 2020; Vasconcelos et al., 2021; Zhang, 2019; Zou et al., 2020; Lee et al., 2020). In a recent study, Azulay & Weiss (2019) found that an image classifier can change its top-1 prediction if the image is translated by only one pixel. Their results showed that after translating an image by one pixel, the classifier made a different top-1 prediction for 30% of the 1000 validation images. Zhang (2019) introduced anti-aliasing filters into a deep neural network to make the feature extraction network shift-equivariant, which in term makes the whole network shift-invariant for the down streaming tasks. Compared to these works, our problem is different in that 1) a perceptual similarity metric takes two images as input instead of working on a single input image, and 2) only one of the two images is shifted, thus introducing imperceptible misalignment instead of shifting the two images simultaneously.

This paper aims to develop a shift-tolerant perceptual similarity metric that correlates well with the human judgement on the similarity between images while being robust against imperceptible misalignment between them. We build our metric upon LPIPS, a deep neural network-based metric that is now widely adopted for its close correlation with the human perception. We investigate a variety of elements that can be incorporated into a deep neural network to make it resistant to an imperceptible misalignment. These elements include anti-aliasing filters, striding, pooling, padding, placement of anti aliasing, etc. Based on our findings on these elements, we develop a shift-tolerant perceptual similarity metric that not only is more consistent with human perception but also is significantly more resistant to imperceptible misalignment between a pair of images than existing metrics.

In the remainder of this paper, we first report our study that verifies that viewers are not sensitive to small amount of shifts between two images when comparing them, in Section 3. We then benchmark existing visual similarity metrics and show that these metrics are sensitive to imperceptible shifts between a pair of images in Section 4. We then study several important elements that make a deep neural network-based similarity metric both tolerant to imperceptible shifts and consistent with the human perception of visual similarity in Section 5. We finally report our experiments that thoroughly evaluate our new perceptual similarity metric by comparing it to state of the art metrics and through detailed ablation studies in Section 6.

## 2 RELATED WORK

Visual similarity metrics are commonly used to compare two images or evaluate the performance of many image and video processing, editing and synthesis algorithms. While there are already many established metrics for these tasks, such as PSNR, MSE, SSIM and its variations (Wang et al., 2004; 2003; Wang & Simoncelli, 2005), there is still a gap between their prediction and the human's judgement. This section provides a brief overview of the recent advances in learned perceptual similarity metrics that aim to bridge the gap mentioned above.

In their influential work, Zhang et al. (2018) reported that features from a deep neural network can be used to measure the similarity between two images that is more consistent with the human perception than other commonly used metrics. Accordingly, they developed LPIPS, a perceptual metric learned from a large collection of labelled data. Specifically, LPIPS uses a pre-trained network for image classification tasks or learns a neural network to compute the features for each of the two images or patches, and also learns to aggregate the feature distances into a similarity score. Since its debut, LPIPS has been widely used as a perceptual quality metric. On a related note, the computer vision and graphics community also calculate the difference between the deep features of two images as a loss function to train deep neural networks for image enhancement and synthesis. Such a loss function, often called perceptual loss, enables the neural networks to learn to generate perceptually pleasing images (Dosovitskiy & Brox, 2016; Johnson et al., 2016; Ledig et al., 2016; Niklaus et al., 2017; Sajjadi et al., 2016; Zhu et al., 2016).

Kettunen et al. (2019) developed the E-LPIPS metric that adopts the LPIPS network and uses randomly transformed samples to calculate expected LPIPS distance over them. They showed that E-LPIPS is robust against the Expectation Over Transformation attack (Athalye et al., 2018). Different

from LPIPS, Prashnani et al. (2018) use the differences between features to generate patch-wise errors and corresponding weights, via two different fully-connected networks. Their final similarity score is a weighted average of the patch-wise distances. Czolbe et al. (2020) developed a similarity metric based on Watson's perceptual model (Watson, 1993), by replacing discrete cosine transform with discrete fourier transform (DFT). They posit that their metric is robust against small translations and is sensitive to large translations. Czolbe et al. (2020) used Watson-DFT as a differentiable loss function for image generation via variational autoencoders (Kingma & Welling, 2013).

In earlier work, Wang & Simoncelli (2005) improved SSIM (Wang et al., 2004) by replacing the spatial correlation measures with phase correlations in wavelet subbands which made the metric less sensitive to geometric transformations. Ma et al. (2018) developed a geometric transformation invariant method (GTI-CNN). Our work is closely related to theirs, as GTI-CNN is a similarity metric that is invariant to the misalignment between a pair of images. In their method, Ma et al. (2018) train a fully convolutional neural network to extract deep features from each image and calculate the mean squared error between them as their final similarity. They showed that training the fully convolutional neural network directly on aligned samples leads to a metric that is sensitive to the misalignment, which is consistent with what we found in our study. They reported that augmenting the training samples with small misalignment can make the learned metric significantly more resistant to the misalignment. Compared to this method, our work focuses on designing a deep neural network architecture that is robust to misalignment without any data augmentation. Bhardwaj et al. (2020) followed the understanding of the physiology of the human visual system and developed a fully convolutional neural network that generates a multi-scale probabilistic representation of an input image and then calculates the symmetric Kullback–Leibler divergences between such representations of two images to measure their similarity. They found that such a similarity metric is robust against small shifts between a pair of images. While benchmarking existing metrics, our study also finds that their metric is most robust against small shifts among all the metrics we tested. We posit that the robustness of their method partially comes from training their metric on neighboring video frames that might already have small shifts among them, thus effectively serving as data augmentation, as done by Ma et al. (2018). We consider these as orthogonal efforts in developing a robust similarity metric. Also, as shown in our study, our metric is more consistent with the human judgement and more robust against imperceptible misalignment than these methods, even though our metric is trained on aligned samples directly without any data augmentation.

Our work is most related to deep image structure and texture similarity (DISTS) metric by Ding et al. (2020). They used global feature aggregation to make DISTS robust against mild geometric transformations. They also replaced the max pooling layers with $l_2$ pooling layers (Hénaff & Simoncelli, 2016) in their VGG backbone network for anti-aliasing and found that blurring the input with $l_2$ pooling makes their network more robust against small shifts. Gu et al. (2020) found that existing metrics like LPIPS do not perform well with images generated by GAN-based restoration algorithms. They attributed it to the small misalignment between the GAN results and the ground truth. Therefore, they used $l_2$ pooling (Ding et al., 2020; Hénaff & Simoncelli, 2016) and *BlurPool* (Zhang, 2019) to improve LPIPS. They found that both can improve LPIPS while *BlurPool* performs better. Compared to these two recent papers, our paper systematically investigates a broad range of neural network elements besides *BlurPool*. By integrating these elements together, we develop a perceptual similarity metric that is both robust against small shifts and is consistent with the human visual similarity judgement. Our method outperforms existing metrics, and a variety of recently developed learned metrics. Integrating multiple network elements together makes our metric better than individual ones, including *BlurPool*.

## 3    HUMAN PERCEPTION OF SMALL SHIFTS

As commonly expected, shifting one image by a few pixels will not alter human similarity judgement on a pair of images (Bhardwaj et al., 2020; Xiao et al., 2018). We conducted a user study to verify this common belief. Our hypothesis is that *it is difficult for people to detect a small shift in images*.

In our study, we randomly picked 50 images from the MS-COCO test dataset (Lin et al., 2014). For each participant, we randomly divided these 50 images into 10 groups, each with 5 images. For each image in Group $n$ with $n \in [0, 10)$, we cropped a $256 \times 256$ patch as a reference image and shifted the cropping window by $n$ pixels to produce its shifted version. In this way, we generated

50 pairs of images for each participant. In the end, since we always cropped the reference from the same location, we had an n-pixel shifted version for each of the 50 images, and thus in total, we have 500 pairs of images in our study. During the study, each participant was presented with a pair of images one at a time. The order of the 50 pairs of images is also randomized for each participant. In each trial, a pair of images were placed side by side. The position of the reference image, e.g., right or left, is randomized to avoid any bias. We asked our participants to judge whether a pair of images are the same or not. We refer the reader to Appendix A.1 for more details.

We report the user responses in Figure 2. When the amount of shift is small, participants find it difficult to detect the shift. For pairs of images with the 1-pixel shift and 2-pixel shift, they were considered the same in 80.7% and 56.0% of the responses, respectively. As expected, the shifts become easier to detect as the size of the shifts increases. But even for pairs of the 5-pixel shift, they were still not identified in 26.7% of the responses. As shown in our study, even after being informed about the possible shifts, participants still had difficulty in detecting small shifts. This verifies our hypothesis that it is difficult for people to detect a small shift in images. In addition,

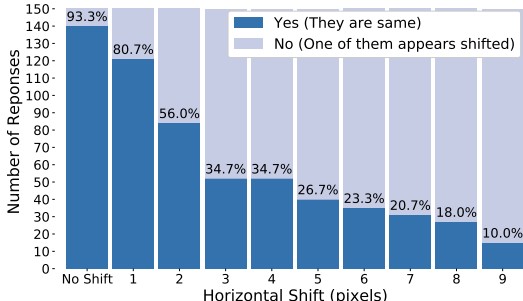

Figure 2: Human perception of small shifts. Image pairs with 1- and 2-pixel shift are deemed the same in 80.7% and 56.0% of the responses, resp.

we use this data and test the consistency of various metrics with the sensitivity of human perception to pixel shifts in Appendix A.1.1. The results of this test provide further evidence that our metric is more consistent in regards to this.

## 4 EFFECT OF SMALL SHIFTS ON SIMILARITY METRICS

To understand how existing similarity metrics handle small shifts between a pair of images, we benchmarked representative metrics, including off-the-shelf metrics, such as L2 and SSIM, and recent deep learning-based metrics. We derived a new dataset from the Berkeley-Adobe Perceptual Patch Similarity Dataset (BAPPS) (Zhang et al., 2018) in our study. The original BAPPS dataset consists of 36,344 examples, each with a reference image $I_r$, and two distorted images $I_1$ and $I_2$. These samples cover a wide range of common distortions, including traditional, CNN-based, and from real-algorithms such as superresolution, frame interpolation, deblurring, and colorization. Please refer to Zhang et al. (2018) for more details. For each sample in the BAPPS dataset, we shifted the distorted images horizontally by $k$ pixels where $k \in \{1, 2, 3\}$. To avoid any boundary artifacts from shifting, we cropped each shifted image $I_i$ as follows.

$$\hat{I}_i = I_i[\, 0:h, \ k:(w+k-3)] \tag{1}$$

Table 1: Accuracy (2AFC) & shift-tolerance ($r_{rf}$) of various metrics on the BAPPS validation dataset. 2AFC is computed on the BAPPS dataset resized $64 \times 64$ while $r_{rf}$ scores are obtained from its shifted version of size $64 \times 61$.

| Network | 2AFC | $r_{rf}$ | | |
|---|---|---|---|---|
| | | 1 pixel | 2 pixel | 3 pixel |
| L2 | 62.91 | 12.27 | 23.07 | 28.83 |
| SSIM (Wang et al., 2004) | 63.08 | 13.08 | 25.50 | 32.74 |
| CW-SSIM (Wang & Simoncelli, 2005) | 60.55 | 11.33 | 18.28 | 23.22 |
| E-LPIPS (Kettunen et al., 2019) | 69.23 | 8.72 | 10.67 | 12.34 |
| GTI-CNN (Ma et al., 2018) | 63.74 | 9.37 | 12.32 | 16.25 |
| DISTS (Ding et al., 2020) | 68.89 | 5.57 | 8.20 | 10.07 |
| PIM-1 (Bhardwaj et al., 2020) | 69.45 | **1.63** | **3.06** | **4.39** |
| PIM-5 (Bhardwaj et al., 2020) | 69.47 | 2.28 | 3.56 | 5.19 |
| LPIPS (Alex) (Zhang et al., 2018) | 69.83 | 6.79 | 8.90 | 9.70 |
| LPIPS (Alex) retrained from scratch[*†] | **70.04** | 9.25 | 9.34 | 11.55 |
| LPIPS (Alex) ours[*†] | 69.83 | 3.48 | 4.75 | 6.84 |

[*] Trained on image patches of size 64 using author's ([†]) setup.

where $w$ and $h$ are the original image size. In this way, all the images in our test were of size $(w-3) \times h$ without regard to the amount of shift, which eliminates the effect of image sizes when we test how the amount of shift affects the performance of metrics. The reference images were also cropped to the same size as the distorted images but no shifts were applied. In addition, we also cropped all the images in each original sample to the size of $(w-3) \times h$ in order to make the shifted sample and the original sample the same size to avoid the effect of the image size on a similarity metric in our late experiments. No shift was introduced to the original samples. A 3-pixel shift in our setting is equivalent to shifting 1.2% of the pixels for the images of size 256 x 256 pixels.

When evaluating a similarity metric, we applied it to both the original sample in the BAPPS dataset as well as its corresponding shifted once. Specifically, for each sample, we obtained two pairs of similarity scores, $(s_1, s_2)$ and $\hat{s}_1, \hat{s}_2$. $(s_1, s_2)$ are the similarity scores between $I_1$ and its reference

image $I_r$, and $I_2$ and $I_r$, respectively. $(\hat{s}_1, \hat{s}_2)$ are the corresponding pair of similarity scores for the shifted sample. Each pair of scores indicates which of the two distorted images is more similar to the reference image according to the metric used in the test. We count the number of samples for which the similarity rank flips when a sample was shifted and compute the rank-flip rate as follows.

$$r_{rf} = \frac{1}{N} \sum_{l=1}^{N} (s_1^l < s_2^l) \neq (\hat{s}_1^l < \hat{s}_2^l) \quad (2)$$

where $r_{rf}$ is the rank-flip rate and $N$ is the number of samples in the test set. We use $r_{rf}$ to evaluate how robust a metric is against the small shift between a pair of images.

For all the learned metrics involved in this study, we used the trained models shared by their authors unless otherwise noted. While the image size of the BAPPS dataset is $256 \times 256$, some models shared by their authors were trained on $64 \times 64$ resized images. Therefore we conducted studies on these two sizes separately to provide fair and informative comparisons.

Table 2: Accuracy (2AFC) & shift-tolerance ($r_{rf}$) across various metrics on the BAPPS validation dataset. 2AFC is computed on the BAPPS dataset of original size $256 \times 256$ while $r_{rf}$ is obtained from its shifted version of size $256 \times 253$.

| Network | 2AFC | $r_{rf}$ | | |
|---|---|---|---|---|
| | | 1 pixel | 2 pixel | 3 pixel |
| L2 | 62.92 | 3.59 | 7.55 | 10.82 |
| SSIM (Wang et al., 2004) | 61.41 | 3.16 | 7.20 | 13.73 |
| CW-SSIM (Wang & Simoncelli, 2005) | 61.48 | 3.91 | 6.88 | 9.47 |
| MS-SSIM (Wang et al., 2003) | 62.54 | 2.22 | 5.83 | 10.66 |
| PIEAPP Sparse (Prashnani et al., 2018) | 64.20 | 2.83 | 3.19 | 3.81 |
| PIEAPP Dense (Prashnani et al., 2018) | 64.15 | 2.97 | 1.37 | 3.33 |
| PIM-1 (Bhardwaj et al., 2020) | 67.45 | 0.79 | 1.70 | 2.52 |
| PIM-5 (Bhardwaj et al., 2020) | 67.38 | 1.01 | 1.88 | 2.96 |
| GTI-CNN (Ma et al., 2018) | 63.87 | 3.95 | 4.91 | 7.88 |
| DISTS (Ding et al., 2020) | 68.83 | 2.85 | 2.89 | 4.03 |
| E-LPIPS (Kettunen et al., 2019) | 68.22 | 5.84 | 5.86 | 5.77 |
| LPIPS (Alex) (Zhang et al., 2018) | 68.59 | 2.81 | 3.41 | 3.84 |
| LPIPS (Alex) retrained from scratch*† | 70.54 | 2.58 | 3.59 | 3.53 |
| LPIPS (Alex) ours*† | 70.39 | 0.66 | 1.24 | 1.79 |
| LPIPS (Alex) retrained from scratch*‡ | **70.65** | 2.87 | 3.92 | 3.74 |
| LPIPS (Alex) ours*‡ | 70.48 | **0.57** | **1.06** | **1.50** |

∗ Trained on patches of size 256 using author's (†) / our (‡) setup.

We report the results in Tables 1 and 2. All scores are obtained by averaging over examples in each distortion category in the BAPPS dataset and then averaging over all the categories. The two-alternative forced choice (2AFC) scores were obtained from the original BAPPS dataset that indicates how a metric's prediction correlates with the human opinion (Zhang et al., 2018). The rank-flip rate ($r_{rf}$) is calculated from the shifted dataset. It shows how robust a metric is to the shift between a distorted image and its reference image. As reported in Table 2, the learned metrics match the human perception better than the non-learned ones such as L2, SSIM, and MS-SSIM. However, even these learned metrics are sensitive to small shifts except for the recent metric, PIM (Bhardwaj et al., 2020). Compared to these existing metrics except PIM, our metrics are more consistent with human perception as per 2AFC scores and more robust against small shifts. Overall, our method is comparable to PIM. Our method outperforms PIM on images of size $256 \times 256$ (Table 2) but does not work as well as it on smaller images (Table 1). As discussed in Section 2, PIM is trained from neighboring video frames which often contain small shifts, which makes it robust against the imperceptible shifts. Our work is orthogonal to PIM in that we investigate neural network elements to build a robust similarity metric. Therefore, we purposely trained our metrics on the BAPPS dataset without any data augmentation.

## 5 ELEMENTS OF SHIFT-TOLERANT METRICS

Some recent papers reported that training a deep neural network using samples with shifted images through either data augmentation or neighboring video frames can make a learned similarity metric robust against small shifts between a pair of images (Ma et al., 2018; Bhardwaj et al., 2020). This paper aims to solve this problem from a different perspective; we investigate how one can design a deep neural network that can be resistant to small shifts. We select the LPIPS network architecture as our baseline framework as it correlates with the human visual similarity judgment well (Zhang et al., 2018). To make this paper self-complete, we briefly describe the LPIPS framework. As illustrated in Figure 3, LPIPS first uses a backbone network, such as AlexNet (Krizhevsky et al., 2012) and VGG (Simonyan & Zisserman, 2015), to extract multi-level

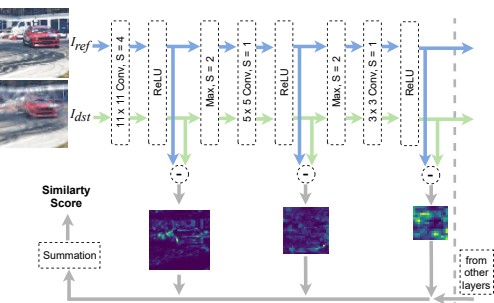

Figure 3: LPIPS framework. The same feature extraction network (AlexNet) is used to extract feature embeddings from $I_{dst}$ and $I_{ref}$. The difference between these embeddings is calculated at different levels and is combined together as the similarity between $I_{dst}$ and $I_{ref}$.

feature embeddings from a distorted image $I_{dst}$ or its reference image $I_{ref}$. We denote the resulting feature embeddings as $F_{dst}$ and $F_{ref}$, respectively. It then calculates the difference between $F_{dst}$ and $F_{ref}$ at all the levels and linearly combines the embedding difference at different levels into a final similarity / difference score, denoted as $d(F_{dst}, F_{ref})$. The combination coefficients and the feature extraction network are learned or fine-tuned.

Below we discuss how various neural network elements affect a similarity metric and how they can be improved to handle imperceptible shifts between a pair of images. Our focus is to develop a feature extraction network to generate feature embeddings from a pair of images that 1) are invariant to imperceptible shifts and 2) lead to a metric that correlates well with the human judgements.

**Reducing Stride.** Striding is widely used in a deep neural network to reduce the input size. For instance, AlexNet has a strided convolution (stride=4) in its first convolutional layer (*conv-1*) and many max pooling operators with stride=2 in the rest of the network. However, it is commonly known that striding with size >1 leads to the sampling rate falling well below the Nyquist rate, which causes aliasing artifacts. In their experiments with image classification tasks, Azulay & Weiss (2019) showed that AlexNet without any subsampling is significantly less sensitive to translations and also maintains its accuracy. Similarly, we also investigate the reduction of the stride size in the convolutional layers in the LPIPS framework to make it more resistant to imperceptible shifts at no expense of its consistency with the human visual similarity perception.

**Anti-aliasing.** Convolution is the most common operator for a deep convolutional neural network. A pure convolutional operator is shift-equivariant instead of being shift-invariant (Nair & Hinton, 2010). Shift equivariance makes a learned similarity metric sensitive to small shifts as small shifts between two images $I_{dst}$ and $I_{ref}$ will be transferred to the shifts between their feature embeddings $F_{dst}$ and $F_{ref}$, which will in term drastically increase the distance between the feature embeddings $d(F_{dst}, F_{ref})$ as shown in Figure 4 (b). Downsampling in a neural network improves its shift invariance. Typically, downsampling can be achieved by a strided convolutional operator or a strided pooling operator with stride $n$ ($n > 1$). However, as discussed earlier in Reducing Stride, striding introduces aliasing. While reducing stride size lessens aliasing, it prevents the network from reducing the feature size.

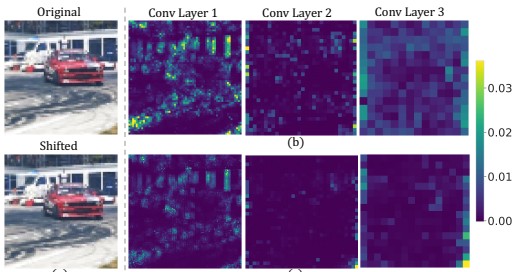

Figure 4: Feature embedding difference maps at different levels. (a) an input image and its one-pixel shifted version. (b) difference maps between embeddings extracted by the original AlexNet. (c) difference maps between embeddings extracted by AlexNet augmented with anti-aliased strided convolution and pooling layers.

To keep the benefit of downsampling while reducing stride, Zhang (2019) invented a *BlurPool* operator. Take max pooling with stride $n$ as an example. Such a max pooling operator can be decomposed into two steps: max pooling with stride 1, followed by a downsampling operator with stride $n$. To reduce the aliasing artifacts, Zhang (2019) followed the pre-filtering idea for anti-aliasing and replaced this max pooling operator with a sequence of three operators: a max pooling with stride 1, a Gaussian filter, and a downsampling operator with stride $n$. The last two operators are combined into as a single operator, called *BlurPool*. Similarly, a convolution operator with stride $n$ can be replaced with its anti-aliased version as a convolution operator with stride $b$ and *BlurPool* with stride $n/b$. Zhang (2019) found that replacing the original convolutional and pooling layers in a feature extraction neural network with their *BlurPool* versions helps generate feature embeddings that make the downstreaming tasks more shift invariant. *BlurPool* uses a fixed Gaussian filter for blurring and may lose some spatial features that are important attributes defining the quality of an image. Zou et al. (2020) developed an adaptive anti-aliasing filter by learning a low-pass filter that is more content-aware. In this paper, we replace the strided convolution layers or pooling layers in the LPIPS framework with *BlurPool* or adaptive anti-aliasing filters to make it invariant to imperceptible shifts among images in a pair. Figure 4 (c) shows that while anti-aliased convolution and pooling layers cannot make the feature network completely shift-invariant, they significantly reduce the difference between the feature embeddings from a pair of shifted images.

**Location of Anti-aliasing.** In a deep neural network, such as AlexNet used in LPIPS, a convolution layer is usually followed by an activation function like *ReLU*. According to Zhang (2019), the activa-

tion function is inserted between the stride-reduced convolutional layer and *BlurPool*, as illustrated in Figure 5 (a). Vasconcelos et al. (2021) created variants of the anti-aliased strided convolution by placing the anti-aliasing filter at different locations, specifically, before or after the convolution operation. They found that some variants can lead to stronger learned inductive priors. But, will they provide significant improvements in shift tolerance? We build upon their findings and design variations of the anti-aliased strided convolutions. Specifically, we modify AlexNet *conv-1* as illustrated in Figure 5 and explain the variants below.

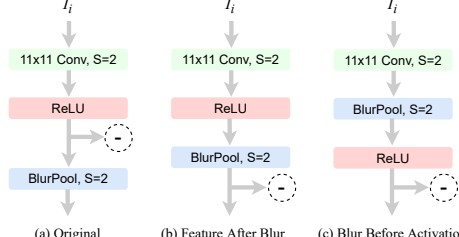

*Original.* As shown in Figure 5 (a), we follow the original design of *BlurPool* and put it after *ReLU* (Zhang, 2019). For anti-aliasing, the stride size of *conv-1* is reduced from 4 to 2 and the *BlurPool* layer has a stride of 2 so that the total stride of 4 is preserved in this anti-aliased version. We take the output of *ReLU* as the feature embedding to calculate the similarity.

Figure 5: Alternative positions of *BlurPool*.

*Feature after blur.* In the above design, the feature embedding is used before *BlurPool*. This effectively reduces the anti-aliasing effect on the feature embeddings although the reduced stride size in *conv-1* still offers some level of anti-aliasing. Therefore, we investigated a variation of the anti-aliased convolution by taking the output of *BlurPool* as the feature embedding to be used for similarity calculation, as illustrated in Figure 5 (b).

*Blur before activation.* Vasconcelos et al. (2021) suggested that blurring after the non-linearity, as done in Figure 5 (a) and (b), prevents high frequency from getting passed on to subsequent layers. Following their findings, we adopted their design by placing *BlurPool* before *ReLU* to keep the high-frequency information from *ReLU*, as shown in Figure 5 (c).

**Border Handling.** Islam et al. (2020) reported that feature embeddings extracted by a convolutional neural network encode absolute position information. This has an important implication for a learning-based similarity metric that feature embeddings from a convolutional neural network are position-dependent and are not shift-invariant. They found that zero padding can relieve this boundary problem for computer vision tasks that are sensitive to spatial information. Kayhan & Gemert (2020) further proposed the concept of full convolution (F-Conv), in which every element of the filter needs to be applied to every pixel in the input image. They implemented F-Conv as a regular convolutional operator with zero padding of $2k$ where $2k + 1$ is the filter kernel size, as illustrated in Figure 7 (Appendix A.2). Note, F-Conv will make the output of an un-strided convolution operator $2k$ larger than the input. They reported that F-Conv is least sensitive to the absolute position of the objects for image classification tasks. Inspired by these works, we replace the regular convolution operators with F-Conv in the LPIPS framework and increase the padding size in *BlurPool* operators to achieve better shift-invariance.

**Pooling.** Max pooling is well known for being more shift invariant than average pooling. We investigate whether its anti-aliased version, *MaxBlurPool* (described earlier in Anti-aliasing) is also more shift invariant than *AvgBlurPool*, the anti-aliased version of average pooling when used in the LPIPS framework. Average pooling in its original form already supports anti-aliasing. We follow Zhang (2019) and implement *AvgBlurPool* with a stride of $n$ as Gaussian filtering followed by downsampling with a factor of $n$.

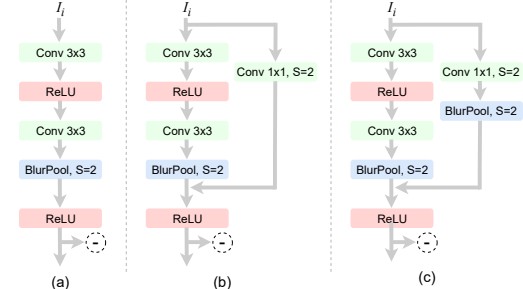

Figure 6: Anti-aliased skipped connection. (a) VGG-like network with AvgBlurPool, (b) with skip connection, and (c) with anti-aliased skip.

**Strided-skip Connections.** Skip connection is widely used in a deep neural network to speedup training and obtain a high quality neural network model. We investigate whether skip connection helps improve shift invariance of a learned similarity metric. As discussed in Vasconcelos et al. (2021), a strided skip connection introduces aliasing for the same reason discussed earlier in Anti-aliasing. We therefore explore anti-aliased strided skip connections, as shown in Figure 6.

## 6 EXPERIMENTS

We built upon the LPIPS framework and incorporated the elements discussed in Section 5 to investigate how these elements help develop a similarity metric that is consistent with the human visual similarity judgement and is robust against imperceptible shifts. We first compare our metrics to both off-the-shelf metrics, such as SSIM and MS-SSIM, and the recent learned similarity metrics. We then conduct ablation studies to evaluate how these elements work.

**Comparisons to Existing Metrics.** In Section 4, we derived a shifted dataset from the BAPPS dataset and compared our metrics to representative existing metrics (Bhardwaj et al., 2020; Ding et al., 2020; Ma et al., 2018; Kettunen et al., 2019; Prashnani et al., 2018; Wang et al., 2004; 2003; Zhang et al., 2018). In our experiments, we adopt the 2AFC metric to evaluate how consistent a metric is with human judgment, and the rank-flipping rate, $r_{rf}$, to evaluate how robust it is against small shifts. As shown in Table 1 and 2, our metrics are both more consistent with human visual similarity judgment and more robust against imperceptible shifts than most of them, except a recent metric PIM (Bhardwaj et al., 2020), to which our method is comparable. PIM achieves shift robustness by training on neighboring video frames that often have small shifts. We work on an orthogonal solution by investigating neural network elements to make the learned metric robust and therefore only train our metrics on the examples without any shift through data augmentation. We further evaluated the metrics on the perceptual validation dataset from the Challenge on Learned Image Compression (CLIC, 2021). The results in Table 3 are consistent with previous results, i.e., our method outperforms all the other methods in terms of accuracy and is more shift-robust than others except PIM, which is similarly robust to ours.

Table 3: Experiments on the CLIC dataset.

| Network | Accuracy(%) | Number of rank flips | | |
|---|---|---|---|---|
| | | 1 pix | 2 pix | 3 pix |
| L2 | 58.16 | 833 | 2102 | 2214 |
| SSIM (Wang et al., 2004) | 60.00 | 349 | 931 | 1109 |
| PIEAPP (Prashnani et al., 2018) | 75.44 | 91 | 134 | 158 |
| E-LPIPS (Kettunen et al., 2019) | 74.44 | 212 | 251 | 317 |
| DISTS (Ding et al., 2020) | 75.63 | 28 | 36 | 50 |
| PIM-1 (Bhardwaj et al., 2020) | 73.79 | **13** | 22 | 33 |
| LPIPS(Alex) (Zhang et al., 2018) | 73.68 | 90 | 108 | 121 |
| LPIPS(Alex) retrained from scratch[*][†] | 76.53 | 59 | 51 | 62 |
| LPIPS(Alex) ours[*][†] | **76.97** | 17 | **14** | **21** |

[*] Trained on image patches of size 64 using author's ([†]) setup.

**Ablation Studies.** We now examine how individual network elements affect our metrics. In these studies, we trained all our metrics using the original BAPPS training set on their original size of $256 \times 256$. We purposely did not train on the shifted version to focus on neural network element designs. To train our metrics, we used the loss function: $MSE(s, h)$, where $s = s_1/(s_1 + s_2)$, $s_1$ and $s_2$ are the predicted similarity scores of two images $I_1$ and $I_2$ to their corresponding reference image, and $h$ is the human score. We trained our metrics using the same settings as Zhang et al. (2018) except we used a lower dropout rate of 0.01. We tested all our metrics on the shifted testing dataset to obtain the rank-flipping rate. To obtain the 2AFC scores, we ran our metrics on the full-size images (with no shift) of the original BAPPS dataset so that we could verify whether our metrics sacrifice consistency with human visual similarity judgment to be robust against imperceptible shifts.

We first examine elements discussed in Section 5 individually. We use AlexNet as the backbone feature extraction network with the LPIPS framework as it provides the best result among other backbone networks (Zhang et al., 2018). As reported in Table 4, anti-aliasing via *BlurPool* can greatly improve LPIPS's robustness against imperceptible shifts. Reducing stride size in its strided convolutional layer (*conv-1*) also helps making it significantly more robust at little expense of the 2AFC score. Combining *BlurPool* with reducing stride size makes the network even more robust against imperceptible shifts and more consistent with human judgment based on the 2AFC score. Then a larger reflection padding size also helps as it reduces the position information encoded in the feature embeddings from the image boundaries, as discussed in Section 5. However, F-Conv, also designed to reduce the boundary issue, does not help. While the learned *BlurPool* (Zou et al., 2020) also helps, it is not as effective as the original version for our task of making a robust similarity metric.

Table 4: Effect of (1) anti-aliasing (AA) via *BlurPool*, (2) F-Conv, (3) reduced stride, & (4) adaptive-AA[§] on learned metrics.

| AA (BlurPool) Reflection-Pad | | F-Conv | Stride in *conv-1* | 2AFC | $r_{rf}$ | | |
|---|---|---|---|---|---|---|---|
| 1 | 2 | | | | 1 pixel | 2 pixel | 3 pixel |
| | | | 4 | 70.65 | 2.87 | 3.92 | 3.74 |
| ✓ | | | 2 | 70.53 | 1.85 | 2.22 | 2.58 |
| | ✓ | | 2 | **70.67** | 1.46 | 1.82 | 2.25 |
| | | ✓ | 4 | 70.57 | 2.78 | 3.92 | 3.91 |
| ✓ | | ✓ | 2 | 70.52 | 1.77 | 2.15 | 2.48 |
| | | | 2 | 70.54 | 1.84 | 2.28 | 2.34 |
| | ✓ | | 1 | 70.42 | 0.66 | **1.13** | 1.83 |
| | ✓ | ✓ | 1 | 70.44 | **0.63** | 1.14 | **1.68** |
| ✓[§] | | | 2 | 70.57 | 2.63 | 3.36 | 3.16 |
| | ✓[§] | | 2 | 70.63 | 2.80 | 3.57 | 3.39 |
| | ✓[§] | ✓ | 2 | 70.52 | 2.95 | 4.13 | 3.93 |

[§] (Zou et al., 2020)

We test on different backbone feature networks, including VGG-16 (Simonyan & Zisserman, 2015), ResNet-18 (He et al., 2016), and SqueezeNet (Iandola et al., 2016). While reducing the stride size is effective, not all networks have a strided convolution layer. Hence, we focus on *BlurPool* applied to pooling layers. As shown in Table 5, *BlurPool* significantly improves the robustness of other backbone networks as well. What is interesting is the effect of the padding size within these backbone networks. While a larger padding size improves 2AFC scores, it does not make shift-invariance better for VGG-16 and ResNet-18. For them, the anti-aliased convolution layers have a stride size of 1, which leads to minor boundary issues. We conjecture that this makes a larger padding size unnecessary.

Table 5: Anti-aliasing via *BlurPool* can significantly improve shift-tolerance and often improve 2AFC scores consistently for different backbone feature extraction networks.

| Network | AA (BlurPool) Reflection-Pad | | 2AFC | $r_{rf}$ | | |
|---|---|---|---|---|---|---|
| | 1 | 2 | | 1 pixel | 2 pixel | 3 pixel |
| **VGG-16** | | | 70.03 | 3.01 | 3.76 | 3.44 |
| | ✓ | | 70.05 | **0.66** | **1.08** | **1.44** |
| | | ✓ | **70.07** | **0.66** | 1.12 | 1.82 |
| **ResNet-18** | | | 69.86 | 2.67 | 3.35 | 3.77 |
| | ✓ | | 69.95 | **0.82** | **1.51** | **2.19** |
| | | ✓ | **70.14** | 1.07 | 1.81 | 2.38 |
| **Squeeze** | | | 69.61 | 7.41 | 7.58 | 10.35 |
| | ✓ | | 69.24 | **2.03** | 3.06 | 3.93 |
| | | ✓ | **69.44** | 2.10 | **2.48** | **3.42** |

We also examine the effect of the location of *BlurPool* within AlexNet. As reported in Table 6, the original version (Figure 5 (a)) works best when the stride size is 2 in *conv-1*. With a smaller stride size, it does not work as well as Blur Before Activation (Figure 5 (c)). This is in part consistent with what was found by Vasconcelos et al. (2021). In the original design,

Table 6: Effect of *BlurPool* locations within an anti-aliased strided convolution as illustrated in Figure 5.

| Anti-Alias (BlurPool) in *Conv-1* | Stride | BlurPool Location | 2AFC | $r_{rf}$ | | |
|---|---|---|---|---|---|---|
| | | | | 1 pixel | 2 pixel | 3 pixel |
| ✓ | 2 | Original | **70.67** | **1.46** | **1.82** | **2.25** |
| ✓ | 2 | FeatureAfterBlur | 70.55 | 1.73 | 1.84 | 2.49 |
| ✓ | 2 | BlurBeforeActivation | 70.50 | 2.06 | 2.02 | 2.74 |
| ✓ | 1 | Original | 70.42 | 0.66 | 1.13 | 1.83 |
| ✓ | 1 | FeatureAfterBlur | **70.52** | 0.69 | 1.11 | 1.60 |
| ✓ | 1 | BlurBeforeActivation | 70.48 | **0.57** | **1.06** | **1.50** |

*BlurPool* is placed after the activation layer for anti-aliasing at the expense of the reduction of the high-frequency information from the activation layer. With the need for anti-aliasing due to a larger stride size, this trade-off works out. However, when stride size is 1, the need for anti-aliasing is reduced; therefore, it is more helpful to place *BlurPool* before the activation layer to avoid the loss of high-frequency information. Thus, Blur Before Activation works better when the stride size is 1.

Table 7 shows that *MaxBlurPool* has better shift tolerance but lower 2AFC scores (accuracy) than *AvgBlurBool*. Moreover, using anti-aliased strided-skip connections leads to higher accuracy with a negligible drop in shift tolerance.

Table 7: Effects of different pooling layers and anti-aliased strided skip connections.

| Network elements | 2AFC | $r_{rf}$ | | |
|---|---|---|---|---|
| | | 1 pixel | 2 pixel | 3 pixel |
| AvgBlurPool | **69.88** | 1.24 | 1.86 | 2.18 |
| MaxBlurPool | 69.58 | **0.95** | **1.54** | **2.12** |
| AvgBlurPool & Strided Skip | 69.82 | 1.38 | 2.06 | 2.61 |
| AvgBlurPool & Anti-aliased Skip | **70.05** | 1.26 | 1.89 | **2.32** |
| MaxBlurPool & Anti-aliased Skip | 69.86 | **1.07** | **1.66** | 2.35 |

**Summary.** Among the network elements we investigated, anti-aliased strided convolution, anti-aliased pooling, and reduction of stride size are most effective to develop a perceptual similarity metric that is robust against imperceptible shifts. These findings are consistent across a variety of backbone network architectures. A larger padding size helps reduce the position information due to the boundary issues encoded in the feature embeddings. Anti-aliased skip connection can help improve accuracy but with little effect on shift invariance. The position of *BlurPool* matters. It should be placed before the activation layer if its precedent convolution uses a small stride size.

# 7 CONCLUSION

This paper reported our investigation on how to design a deep neural network as a learned perceptual image similarity metric that is both consistent with the human visual similarity judgement and robust against the imperceptible shift among a pair of images. We discussed various neural network elements, such as anti-aliased strided convolution, anti-aliased pooling, the placement of *BlurPool*, stride size, and skip connection and studied their effect on a similarity metric. We found that using anti-aliasing strided convolutions and pooling operators and reducing stride size are very helpful to make a learned similarity metric shift-invariant. Our experiments show that by integrating these elements into a neural network, we are able to develop a learned metric that is more robust against imperceptible shifts and more consistent with the human visual similarity judgement.

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

## A  APPENDIX

### A.1  USER STUDY

We recruited 32 participants for our study. These participants have a wide range of professional background, including computer science, business, medicine, arts and education. Most of them are between 20 to 35 years old.

To ensure the quality of this user study, we removed the responses from two participants who failed to pass a validation test. Specifically, if a participant identified a pair of images with 0-pixel shift as *different* or a pair of images with 9-pixel shift as *the same* for more than two-thirds of the time in the study, we did not include the response from that participant. In total, 30 participants passed our validation test. We obtained responses to 1500 trials in total, with 150 responses for each of the $n$-pixel shifts.

It is important to disclose that before the study, we informed our participants that there might be shifts between some pairs of images to raise their alertness to the potential difference created by the shift. While this might bias participants, we found it helpful to obtain a more informative understanding of the human perception of small shifts; otherwise, participants tended to overly overlook the difference between a pair of images.

In total, we have 500 samples with a pair of images where one of the images is shifted. Each user was presented with 5 samples for each 0-9 pix-shift randomly. We made sure that no user saw the same sample twice in our study. The number of responses to each sample varied and the mean number of responses per sample is 3 and the standard deviation is 1.33.

Why we chose 50 images? We generated 500 pairs of images, with 0-9 pixel shifts from them. To maintain the quality of our study and avoid boring the users, we only presented 50 samples to each user. Interestingly, humans managed to detect the shift for a 2 pixel shift in 50% of cases. We attribute this partially to the fact that the users were informed that there might or might not be a shift between a pair of images. This indeed introduced biases into the study such that their sensitivity to the shifts is likely increased. We chose to do so as, in our pilot study, we found that users were very confused when we asked them if a pair of images looked the same or not. Many of them thought

if we were asking them to compare high-level features such as objects in the two images or if there were some artifacts in one of the pair of images.

Furthermore, we analyzed the variability of the user responses. In our analysis, if a user noticed the shift between a pair of images, we label the response as 1 and 0 otherwise. We then calculate the standard deviation of the user responses for each image. The average (avg.) of the standard deviation (std.) in the responses per sample is 0.2 with a standard deviation of 0.23.

Table 8: Variability in user responses.

| Pixel-shift | Avg. of std. per sample | Std. of std. per sample |
|:---:|:---:|:---:|
| 0 | 0.09 | 0.17 |
| 1 | 0.19 | 0.23 |
| 2 | 0.34 | 0.21 |
| 3 | 0.24 | 0.23 |
| 4 | 0.3 | 0.24 |
| 5 | 0.23 | 0.24 |
| 6 | 0.21 | 0.24 |
| 7 | 0.12 | 0.2 |
| 8 | 0.18 | 0.23 |
| 9 | 0.13 | 0.21 |

Finally, we compute the average standard deviation for the whole group of samples with the same amount of shift and report the results in Table 8. With no or only a 1-pixel shift, users were consistently sure that the images in each pair were the same. Similarly, with a very large shift (6 to 9 pixels), users consistently indicated that the images were shifted. In contrast, we see more variability in user responses when the shift is 2 to 5 pixels. Hence, for images with a 2 to 5-pixel shift, users were doubtful whether images were shifted or not, and their responses had a high variation.

### A.1.1 JUST NOTICEABLE DIFFERENCES (JND)

Table 9: Consistency of perceptual similarity metrics with the sensitivity of human perception to pixel shifts.

| Metric | JND mAP% |
|:---|:---:|
| SSIM (Wang et al., 2004) | 0.722 |
| LPIPS (Alex) (Zhang et al., 2018) | 0.757 |
| LPIPS (Alex) retrained from scratch[*†] | 0.740 |
| LPIPS (Alex) **ours**[*†] | 0.771 |
| LPIPS (VGG) (Zhang et al., 2018) | 0.770 |
| LPIPS (VGG) retrained from scratch[*†] | 0.769 |
| LPIPS (VGG) **ours**[*†] | **0.775** |
| DISTS (Ding et al., 2020) | 0.766 |
| PIM-1 (Bhardwaj et al., 2020) | 0.773 |

[*] Trained on image patches of size 64 using author's ([†]) setup.

We conducted the following test to study how consistent our shift-tolerant perceptual similarity metric is with the human perception results reported in Figure 2. In our study reported in Figure 2, we had asked our participants if the two images, which may be shifted by a few pixels, were the same or not. Using these responses, we perform a just noticeable difference test. We make use of only those samples which have at least 3 human responses. There were 301 such samples, and the mean number of samples per pixel-shift (0 to 9) is 30.1 with a standard deviation of 1.6 (maximum 33 and minimum 28). Following Zhang et al. (2018), we rank the pairs by a perceptual similarity metric and compute the area under the precision/recall curve (mAP) (as used by Everingham et al. (2010) and Zhang et al. (2018)). The results in Table 9 show that our shift-tolerant LPIPS metrics follow the sensitivity of human perception to pixel shifts more accurately than their vanilla versions. The accuracy of PIM-1 and DISTS is comparable to ours.

## A.2 FULL CONVOLUTION

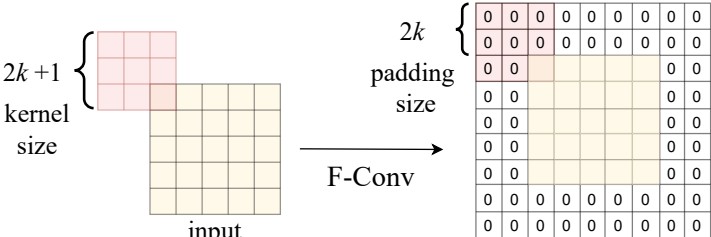

Figure 7: Illustration of Full Convolution (F-Conv), where every value of the filter needs to be applied to each value of an input image. Hence, the input needs to be padded first with padding size $2k$ for a filter with size $2k + 1$ (Kayhan & Gemert, 2020).

## A.3 ADDITIONAL ABLATION STUDIES

We provide a few additional ablation studies to help examine the network elements discussed in our main paper.

### A.3.1 PADDING SIZE

Table 10: Effects of increasing reflection-pad size. This test is conducted for our learned metric using AlexNet as its backbone network.

| AA (*BlurPool*) w/ Reflection-Pad | 2AFC | $r_{rf}$ 1 pixel | 2 pixel | 3 pixel |
|---|---|---|---|---|
| None | 70.53 | 2.11 | 2.51 | 2.58 |
| 1 | 70.53 | 1.85 | 2.22 | 2.58 |
| 2 | **70.67** | **1.46** | **1.82** | **2.25** |

To further examine the effect of the padding size, we added a new variation where no padding is used in all the *BlurBool* layers. As shown in Table 10, padding in the *BlurPool* layers makes our learned metric more robust against small shifts.

### A.3.2 BLUR KERNEL SIZE

Table 11: Effects of increasing the blur kernel size.

| AA (*BlurPool*) Reflection-Pad 1 | 2 | Stride *in Conv-1* | Blue Kernel Size *in Conv-1* | 2AFC | $r_{rf}$ 1 pixel | 2 pixel | 3 pixel |
|---|---|---|---|---|---|---|---|
| ✓ | | 2 | 3 | 70.53 | 1.85 | 2.22 | 2.58 |
| ✓ | | 2 | 5 | **70.60** | **1.57** | **1.67** | **2.21** |
| | ✓ | 2 | 3 | **70.67** | **1.46** | 1.82 | 2.25 |
| | ✓ | 2 | 5 | 70.54 | 1.61 | **1.55** | **2.16** |
| | ✓ | 1 | 3 | 70.42 | **0.66** | **1.13** | 1.83 |
| | ✓ | 1 | 5 | **70.57** | 0.73 | 1.16 | **1.71** |

*BlurPool* is used for anti-aliasing for strided convolution and strided pooling operators (Zhang, 2019). Anti-aliasing is achieved by employing a Gaussian filter to remove the high-frequency in the intermediate result from its previous step of non-striding pooling or convolution operators. A key parameter is the size of Gaussian filter kernel. In our paper, we used a fixed kernel size of 3. We add a new test to examine the effect of the kernel size in *conv-1* on our learned metric. As shown in the first group of Table 11, a large kernel helps anti-aliasing in general and thus makes our metric more shift-tolerant. However, when combined with a reduced stride size (=1), a large kernel can also be counter productive and make our metric less shift-invariant, as shown in the bottom group in Table 11. This is because when the stride size is 1, aliasing is not a big issue. Then a large kernel size removes useful high-frequency information with little gain from anti-aliasing.

### A.3.3 PADDING TYPE

Table 12: Comparing the type of padding, Zero vs. Reflection, in the BlurPool layers.

| AA (*BlurPool*) | | | | $r_{rf}$ | | |
|---|---|---|---|---|---|---|
| Refl-Pad Size 2 | Zero-Pad Size 2 | F-Conv | 2AFC | 1 pixel shift | 2 pixel shift | 3 pixel shift |
| ✓ | | | 70.67 | **1.46** | **1.82** | **2.25** |
| | ✓ | | 70.55 | 2.05 | 2.28 | 2.70 |
| ✓ | | ✓ | 70.52 | 1.77 | 2.15 | 2.48 |
| | ✓ | ✓ | **70.72** | 1.80 | 2.24 | 2.40 |

In all our experiments, we set the default padding type as reflection padding. We now examine if zero padding performs better or not by replacing reflection padding with zero padding in the *BlurPool* layers. As shown in Table 12, we found that reflection padding in general outperforms zero padding. In a recent study, Alsallakh et al. (2021) found reflection pad to be more effective in reducing line artifacts in feature maps. We believe it reduces aliasing that improves shift tolerance.

### A.3.4 ANTI-ALIASING OPERATOR TYPE

Table 13: Comparing BlurPool against $l_2$ pooling.

| Anti-alias | | Blur Kernel Size in all layers | 2AFC | $r_{rf}$ | | |
|---|---|---|---|---|---|---|
| BlurPool | $l_2$ Pooling | | | 1 pix | 2 pix | 3 pix |
| ✓ | | 3 | 70.53 | **1.85** | **2.22** | **2.58** |
| | ✓ | 3 | **70.69** | 2.92 | 3.90 | 3.60 |
| ✓ | | 5 | 70.51 | **1.68** | **1.59** | **2.17** |
| | ✓ | 5 | **70.55** | 1.87 | 1.89 | 2.38 |

We use BlurPool for anti-aliasing in all our experiments. Here we compare the effect of anti-aliasing via *BlurPool* versus $l_2$ *Pooling*. As observed in Table 13, when the blur kernel size is 3, the shift tolerance of the network having *BlurPool* is substantially better than the network with $l_2$ *Pooling* layers, but at the expense of accuracy. This maybe due to a weaker blurring by $l_2$ *Pooling*. As the blur kernel size is increased to 5, the strength of the blurring effect increases in both, and we observe the performance gap reducing, in terms of both accuracy and shift tolerance.

### A.4 RESULTS ON OTHER IQA DATASETS

Table 14: Performance comparison on the TID-2013 (Ponomarenko et al., 2015) dataset.

| Metric | SRCC | KRCC | PLCC |
|---|---|---|---|
| LPIPS (VGG) retrained from scratch[*] | 0.86 | 0.66 | 0.87 |
| LPIPS (VGG) **ours**[*] | 0.86 | 0.67 | 0.87 |
| DISTS (Ding et al., 2020) | 0.83 | 0.64 | 0.85 |

[*] Trained on Kadid-10k.

Table 15: Performance comparison on the LIVE dataset (Sheikh, 2003).

| Metric | SRCC | KRCC | PLCC |
|---|---|---|---|
| LPIPS (VGG) retrained from scratch[*] | 0.94 | 0.0.79 | 0.93 |
| LPIPS (VGG) **ours**[*] | 0.95 | 0.80 | 0.94 |
| DISTS (Ding et al., 2020) | 0.95 | 0.81 | 0.91 |

[*] Trained on Kadid-10k.

For an apples-to-apples comparison with DISTS (Ding et al., 2020), we train the LPIPS metric on the Kadid-10k dataset (Lin et al., 2019). We use the pre-trained model for DISTS that is trained on Kadid-10k and uses a VGG backbone network. As shown in Tables 14 and 15, our shift-tolerant LPIPS metric improves the baseline LPIPS metric and outperforms DISTS on both datasets.

Table 16: Performance comparison on the shifted LIVE dataset (Sheikh, 2003).

| Metric | SRCC | | | |
|---|---|---|---|---|
| | Original | No shift | 1-pixel shift | 2-pixel shift |
| LPIPS (VGG) retrained from scratch[*] | 0.94 | 0.94 | 0.90 | 0.91 |
| LPIPS (VGG) **ours**[*] | 0.95 | 0.95 | 0.95 | 0.94 |
| DISTS (Ding et al., 2020) | 0.95 | 0.95 | 0.95 | 0.95 |

∗ Trained on Kadid-10k.

We conducted a test using the SRCC metric on the shifted version of the LIVE dataset. As reported in Table 16, our upgraded LPIPS metric significantly improves the robustness against small shifts over the baseline LPIPS metric. Our results are comparable to DISTS.

