# OpenReview forum: "Shift-tolerant Perceptual Similarity Metric"
_ICLR.cc/2022/Conference — ICLR 2022 Submitted_

### Official Review · Reviewer_GQvy · 2021-11-02

**Correctness:** 3
**Technical Novelty And Significance:** 1
**Empirical Novelty And Significance:** 3
**Recommendation:** 6
**Confidence:** 5

**Main Review:**

Pros:
1) The paper tackles an important and interesting subject of developing shift-invariant metrics. The authors go one step further and support the claim of the importance of such metrics by conducting an experiment with humans (Section 3 and Figure 1)
2) The authors suggest interesting metrics to measure shift-invariance, which counts the number of samples for which the similarity rank flips when a sample was shifted and computes the rank-flip rate  - eq. 2
3) The authors show that their suggested approach improves the proposed metrics for LPIPS arhcietcture (FIgure 3) and achieves comparable results with SOTA PIM similarity network
4) It is interesting to see influence of the suggested architecture alterntaions on different backbones presented in Table 5

Cons:
1) The experiment with human perception is not clearly explained. First, it is hard to understand the number of different pairs that was eventually evaluated by humans or if there were several humans that were presented with the same pair  -- it would be very helpful to summarize those in a table. Also it is very suprising that humans managed were sensetive to 2 pixel shift in ~ 50% of cases. This surprising fact is only explained in the supplementary material. It might be helpful to revise the paper to allow explanation on specific numbers that were chosen (e.g., why only 50 images) and the included biases (e.g. explaining humans that there are shifts they should look for).
2) The novelty of the paper is limited by minor alternations to the exiting architecture. Those alternations, e.g. anti-aliasing, are not new, as stated by the authors. The authors point our several times "PIM achieves shift robustness by training on neighboring video frames that often have small shifts. We work on an orthogonal solution by investigating neural network elements to make the learned metric robust and therefore only train our metrics on the examples without any shift through data augmentation." , this requires a more elaborative discussion on why we would like to choose this second route (which is also not optimal for a single pixel compared to PIM, as evedient from Tabls 1 and 3) There is no discussion of the effect of these additions on the runtime and comparison of the runtime with PIM.
Table 2 shows that when evaluating bigger images the suggest approach is better than PIM, but the authors do not offer explanation why their approach works better
4) Many experiments yield somewhat not intuitive results, like the padding influence on performance in Table 5. The explanations offered are not very clear. Since the paper does not support the suggested approaches theoretically, it might benefit from more intuitive explanations of the experimental results.


**Summary Of The Paper:**

The paper proposes a few modifications to the existing archiectures for perceptual image similarity that would be robust to the tiny shifts in the image. Specifically, the authors conduct an experiemnt that shows that humans mostly are not sensetive to small shifts of 1 or 2 pixels (Table 1), but most state of the art perceptual image similarity networks are sensetive to such shifts (Table 1 and Table 2). The authors discuss what parts of the network architecture might yield such sensetivity and offer a few modifications (Section 5). In Section 6, the authors perform extensive experiments to check the effectiveness of their suggestions. Finally the authors conclude that using anti-aliasing strided convolutions and pooling operators and reducing stride size are helpful to make a learned similarity metric shift-invariant. Their experiments show that by integrating these elements into a neural network, the learned metric is more robust against imperceptible shifts and more consistent with the human visual similarity judgment  (Table 1,2,3) .

**Summary Of The Review:**

I think the paper has merit, but it will benefit from revision, which empahsizes more motivation on why to choose a suggested approach as opposed to the other leading approach PIM and also why certain empirical results make sense.

---

> ### Author Response · Authors · 2021-11-23
> **Responses to your other concerns**
>
> Thank you for taking the time to review our paper. Below please find our responses.
>
> **2. The authors point our several times "PIM achieves shift robustness by training on neighboring video frames that often have small shifts. We work on an orthogonal solution by investigating neural network elements to make the learned metric robust and therefore only train our metrics on the examples without any shift through data augmentation", this requires a more elaborative discussion on why we would like to choose this second route (which is also not optimal for a single pixel compared to PIM, as evedient from Tabls 1 and 3) There is no discussion of the effect of these additions on the runtime and comparison of the runtime with PIM.**
>
> Re: Data augmentation has been already explored in the past [1] and was useful to make a similarity metric robust to small amounts of distortions like translation, rotation, or blurring. As suggested by the reviwer, we tried an augmentation strategy to make the network robust to translations. We randomly crop out a few pixels from each side before resizing the image patches to square patches of size 64. This gives a similar effect as the pixel shift. We train three variants of LPIPS(Alex) as shown in the table below. The vanilla network trained with augmented data performs better than the vanilla network trained without augmented data, but as we start adding elements that reduce aliasing the additional gains with augmentation diminish. However, further studies should investigate a data augmentation strategy that complements to our metric with the added elements. We will revise our paper to add this test accordingly.
>
> | Metric | **2AFC** | $r_{rf}$ 1pix | $r_{rf}$ 2pix | $r_{rf}$ 3pix |
> |:-----|:-----:|:-----:|:-----:|:-----:|
> | | | | | |
> | LPIPS-Alex | 70.04 | 9.25 | 9.34 | 11.55 |
> | + training with augmented data | 70.35 | 7.22 | 8.97 | 9.63 |
> | ----- ----- ----- ----- ----- | ----- |  ----- |  ----- |  ----- |
> | LPIPS-Alex + Antialiasing | 69.97 | 5.57 | 6.16 | 8.49 |
> | + training with augmented data | 70.26 | 5.22 | 5.74 | 7.87 |
> | ----- ----- ----- ----- ----- | ----- |  ----- |  ----- |  ----- |
> | LPIPS-Alex (ours Shift-tolerant) | 69.83 | 3.48 | 4.75 | 6.84 |
> | + training with augmented data | 69.70 | 3.55 | 4.91 | 6.85 |
>
>
> We measured the inference time on image pairs of size 256 x 256 and report the speed below. This test was conducted using an Nvidia 1080 Ti.
>
> | Model              | Runtime (ms) |
> |--------------------|--------------|
> ||
> | LPIPS(Alex)        | 19.6 |
> | + Antialiasing | 48.1 |
> | + Reducing Stride | 145.8 |
> ||
> | PIM                | 89.7 |
>
> [1] Kede Ma, Zhengfang Duanmu, and Zhou Wang. Geometric transformation invariant image quality assessment using convolutional neural networks. In 2018 IEEE International Conference on Acoustics, Speech and Signal Processing, pp. 6732–6736, 2018.
>
> **3. Table 2 shows that when evaluating bigger images the suggest approach is better than PIM, but the authors do not offer explanation why their approach works better.**
>
> Re: Our conjecture for this is that small images (e.g. 64 x 64) loose too much information for visual similarity assessment, therefore our method works better with large ones (e.g. 256 x 256).
>
>
> **4. Many experiments yield somewhat not intuitive results, like the padding influence on performance in Table 5. The explanations offered are not very clear**
>
> Re: In Table 5, we observe that increasing the reflection pad size in the BlurPool layers increases the 2AFC score (accuracy) across all backbone networks. We attribute this improvement to the reduction in foveation due to an increase in padding size. As discussed in Alsallakh et al., 2020 [2], foveation means an unequal contribution of the input pixels in the CNN feature maps. Foveation here is being caused by an uneven application of the convolution operation at the boundary of feature maps. Hence, increasing the padding size ameliorates the effects of foveation at the corners and edges of the feature maps, helping increase the accuracy. Alsallakh et al., 2020 showed that an incorrect size of input can lead to uneven padding for certain CNN architectures. Uneven padding diminishes the contribution of boundary pixels to subsequent feature maps, i.e., such pixels are less involved in successive computations. Increasing the padding size reduces the foveation in the feature maps and increases the model accuracy.
>
> On the other hand, the effect of the reflection padding size on shift tolerance is not consistent across different backbone networks. We will investigate this furthermore in the future.
>
> [2] Alsallakh, B., Kokhlikyan, N., Miglani, V., Yuan, J. and Reblitz-Richardson, O., 2020, September. Mind the Pad--CNNs Can Develop Blind Spots. In International Conference on Learning Representations.

---

> ### Author Response · Authors · 2021-11-23
> **Responses to your 1st concern**
>
> Thank you for taking the time to review our paper. Below please find our responses.
>
> **1.The experiment with human perception is not clearly explained. First, it is hard to understand the number of different pairs that was eventually evaluated by humans or if there were several humans that were presented with the same pair -- it would be very helpful to summarize those in a table. Also it is very suprising that humans managed were sensetive to 2 pixel shift in ~ 50% of cases. This surprising fact is only explained in the supplementary material. It might be helpful to revise the paper to allow explanation on specific numbers that were chosen (e.g., why only 50 images) and the included biases (e.g. explaining humans that there are shifts they should look for).**
>
> Re: Due to the page limit, we have revised Appendix (A.1) to include more information regarding the user study on how humans perceive small pixel-shifts in images.
>
> In total, we have 500 samples with a pair of images where one of the images is shifted. Each user was presented with 5 samples for each 0-9 pix-shift randomly. We made sure that no user saw the same sample twice in our study. The number of responses to each sample varied and the mean number of responses per sample is 3 and the standard deviation is 1.33.
>
> While we only chose 50 images? We generated 500 pairs of images, with 0-9 pixel shifts from them. To maintain the study quality and avoid boring the users, we only presented 50 samples to each user. As the reviewer pointed out "*suprising that humans managed were sensetive to 2 pixel shift in ~ 50% of cases*". We attribute this partially to the fact that users were informed that there might or might not be shifted between a pair of images. This indeed introduced biases into the study such that their sensitivity to the shifts is likely increased. We chose to do so as, in our pilot study, we found that users were very confused when we asked them if a pair of images looked the same or not. Many of them thought if we were asking them to compare high-level features such as objects in the two images or if there were some artifacts in one of the pair of images.
>
> Furthermore, we also tested if perceptual metrics showcase the same sensitivity towards the pixel-shift as humans do and for this study, we performed a just noticeable difference (JND) test using the data we collected in our user study. As shown in table below, the results show that our metric is more consistent with the sensitivity of human perception to pixel-shifts.
>
> |          **Metric**             |  **JND mAP%** |
> |:-------------------------------|:---------:|
> | SSIM                           |   0.722      |
> | LPIPS-Alex (Zhang et al., 2018) |   0.757     |
> | LPIPS-Alex$^*$                  |   0.740     |
> | LPIPS-Alex **ours**$^*$        |   0.771      |
> | LPIPS-VGG (Zhang et al., 2018) |   0.770     |
> | LPIPS-VGG$^*$                  |   0.769      |
> | LPIPS-VGG **ours**$^*$         |   **0.775**  |
> | DISTS                          |   0.766      |
> | PIM-1                          |   0.773      |
> $*$ Trained on BAPPS image patches of size 64 using author’s setup.

---

### Official Review · Reviewer_LHuY · 2021-11-02

**Correctness:** 4
**Technical Novelty And Significance:** 2
**Empirical Novelty And Significance:** 4
**Recommendation:** 8
**Confidence:** 4

**Main Review:**

### Strengths

* Well-written and easy to follow/understand
* Improving architecture to achieve robustness is a better approach than data augmentation which relies on a longer training (energy consumption)
* Fine analysis of neural network elements (their analysis is useful beyond PSMs)
* Human data

### Weaknesses

* No comparison of their psychometric data and proposed PSM
* (minor) figure and table fonts should be similar to main text font

### Detailed comments

I understand that this is not the authors' main goal, however, their psychometric data could be exploited more to enforce the consistency of PSM with human judgement. The first question is: Do humans all notice the shift in the same images or is there a huge variability ?
Then, I am curious about how PSM could reproduce (if consistent across humans) the sensitivity to shift reported in Figure 2.


**Summary Of The Paper:**

The authors propose to make perceptual similarity metrics (PSM) invariant to small-shifts (few pixels translation) and still consistent with human judgement. To this end, they use an approach based on network architectures to evaluate which elements (anti-aliasing, pooling, striding, padding, skip connection) can achieve shift-invariance.

The paper have multiple contributions:
1. A study on the human perception of small shift estimating their sensitivity
2. A study of the sensitivity of PSMs to small shift
3. A systematic study of neural network architecture elements in relation to shift-invariance
4. A updated/shift-invariant version of the LPIPS metric
5. An ablation study to evaluate which elements contribute or to shift-invariance


**Summary Of The Review:**

This a well-conducted and strong paper which achieve is goal and give interesting insight about neural network architecture elements.

---

> ### Author Response · Authors · 2021-11-23
> **Thank you for the feedback**
>
> We thank the reviewer for the constructive comments. Below please find our responses.
>
> **Do humans all notice the shift in the same images or is there a huge variability?**
>
> Re: As suggested by the reviewer, we analyzed the variability of the user responses. In our analysis, if a user noticed the shift between a pair of images, we label the response as 1 and 0 otherwise. We then calculate the standard deviation of the user responses for each image. Finally, we compute the average standard deviation for the whole group of samples with the same amount of shift and report the results in the table below.
>
> With no or only 1-pixel shift, users were consistently sure that the images in each pair were the same. Similarly, with a very large shift (6 to 9 pixels), users consistently indicated that the images were shifted. In constrast, we see more variability in their responses when the shift is 2 to 5 pixels. Hence, for images with a 2 to 5-pixel shift, users were doubtful whether the images were shifted or not and their responses had a high variation.
>
>
> | Pixel-shift | avg. of std. per sample | std. of std. per sample |
> |:-----------:|:-----------------------:|:-----------------------:|
> |      0      |           0.09          |           0.17          |
> |      1      |           0.19          |           0.23          |
> |      2      |           0.34          |           0.21          |
> |      3      |           0.24          |           0.23          |
> |      4      |           0.3           |           0.24          |
> |      5      |           0.23          |           0.24          |
> |      6      |           0.21          |           0.24          |
> |      7      |           0.12          |           0.2           |
> |      8      |           0.18          |           0.23          |
> |      9      |           0.13          |           0.21          |
>
>
>
> **I am curious about how PSM could reproduce (if consistent across humans) the sensitivity to shift reported in Figure 2.**
>
> Re: As suggested by the reviewer, we conducted the following test to study how consistent our shift-tolerant perceptual similarity metric is with the human perception results reported in Figure 2. In our study reported in Figure 2, we had asked our participants if the two images, which may be shifted by a few pixels, were the same or not. Using these responses, we perform a just noticeable difference test. We make use of only those samples which have at least 3 human responses. There were 301 such samples, and the mean number of samples per pixel-shift (0 to 9) is 30.1 with a standard deviation of 1.6 (maximum 33 and minimum 28). Following Zhang et al. 2018, we rank the pairs by a perceptual similarity metric and compute the area under the precision/recall curve (mAP) (as used in PASCAL VOC 2007). The results in the table below show that our shift-tolerant LPIPS metrics follow the sensitivity of human perception to pixel shifts more accurately than their vanilla versions. The accuracy of PIM-1 and DISTS is comparable to ours.
>
> |          **Metric**             |  **JND mAP%** |
> |:-------------------------------|:---------:|
> | SSIM                           |   0.722      |
> | LPIPS-Alex (Zhang et al., 2018) |   0.757     |
> | LPIPS-Alex$^*$                  |   0.740     |
> | LPIPS-Alex **ours**$^*$        |   0.771      |
> | LPIPS-VGG (Zhang et al., 2018) |   0.770     |
> | LPIPS-VGG$^*$                  |   0.769      |
> | LPIPS-VGG **ours**$^*$         |   **0.775**  |
> | DISTS                          |   0.766      |
> | PIM-1                          |   0.773      |
> $*$ Trained on BAPPS image patches of size 64 using author’s setup.
>
>
> **Figure and Table fonts.**
>
> Re: Thanks for the suggestion. We will make them consistent in our paper.

---

### Official Review · Reviewer_2NrG · 2021-11-05

**Correctness:** 2
**Technical Novelty And Significance:** 2
**Empirical Novelty And Significance:** 2
**Recommendation:** 3
**Confidence:** 5

**Main Review:**

Strengths:

The elements of shift-tolerant metrics in the context of convolution neural networks are well analyzed and validated.

Weaknesses:

1. The reviewer appreciates the authors' effort in summarizing recent work on shift-tolerate IQA (and CNN in general). However, the reviewer believes that it is more important to give oldies but goldies enough credit by citing and describing their work in the current manuscript. Dated back to the era of wavelets, there were researchers, trying to come up with shift-invariant wavelets. In the context of IQA, researchers were thinking about shift-invariant IQA as early as 2005. The reviewer is reluctant to directly point out those references but highly encourages the authors to dig them out.

2. Eq. (2) is not a good measure of shift-invariance in the context of IQA. It is more suitable to think of mild shift as a form of non-structural distortion. When applied to images with severe structural distortions (e.g. JPEG compression), the primary goal of a perceptual metric is to capture such visible structural distortions; and it does not matter whether it is robust to mild shift. Therefore, a more reasonable experimental setup is given in GTI-IQA and DISTS papers: assigning shifted images the same MOSs and computing SRCC/KRCC/PLCC on the whole dataset.

3. Shift-invariance is a small piece story in IQA; the authors are encouraged to show the proposed metric is able to deliver SOTA quality prediction performance on standard IQA datasets such as LIVE, CSIQ, TID-2013, and KADID-10K. And if possible try more image restoration datasets like CLIC.

4. 3 pixels are not sufficient to test the robustness of IQA models to mild shift. A more appropriate hyperparameter setting should be resolution-dependent, e.g., 3% to 5% of pixels are shifted.

5. The authors may contrast their algorithms more to DISTS than LPIPS, and may even build their models on top of DISTS for quality prediction, texture performance, and perceptual optimization reasons. The key to the success of DISTS in fighting mild geometric transformations is global feature aggregation, which is understated by the authors.

6. How is the performance of the proposed method to other mild geometric transformations, such as rotation and dilation?

7. Is the feature mapping of the proposed metric injective (or bijective)?

**Summary Of The Paper:**

A shift-tolerant perceptual similarity metric is proposed based on LPIPS.

**Summary Of The Review:**

See above for the detailed comments.

---

> ### Author Response · Authors · 2021-11-23
> **Responses to other comments**
>
> Thank you for taking the time to review our paper. Below please find our responses.
>
> **There were researchers, trying to come up with shift-invariant wavelets. In the context of IQA, researchers were thinking about shift-invariant IQA as early as 2005.**
>
> Re: We thank the reviewer for pointing this work out to us. We have included a short description of the complex wavelet structural similarity index (CW-SSIM) [1] in the related work section in our revision and added its results in Tables 1 and 2 as well.
>
> [1]  Zhou Wang and Eero P Simoncelli. Translation insensitive image similarity in complex wavelet domain. In IEEE International Conference on Acoustics, Speech, and Signal Processing, volume 2, pp. ii–573, 2005.
>
>
> **The key to the success of DISTS in fighting mild geometric transformations is global feature aggregation, which is understated by the authors.**
>
> Re: Thanks for pointing this out to us. We have revised our paper to more properly discuss DISTS in the last paragraph of Section 2 as follows: "Our work is most related to deep image structure and texture similarity (DISTS) metric by Ding et al. (2020). They used global feature aggregation to make DISTS robust against mild geometric transformations. They also replaced the max pooling layers with l2 pooling layers (H ́enaff & Simoncelli,2016) in their VGG backbone network for anti-aliasing and found that blurring the input with l2 pooling makes their network robust against small shifts."
>
>
> **3 pixels are not sufficient to test the robustness of IQA models to mild shift. A more appropriate hyperparameter setting should be resolution-dependent, e.g., 3% to 5% of pixels are shifted.**
>
> Re: According to our study in Section 3, a shift with 4+ pixels would be easily noticeable. For a pair of images with a noticeable shift, it is debatable whether the shift should be considered when measuring the similarity between them, depending on application scenarios. Our work aims to develop a perceptual similarity metric that is robust against small shifts that are imperceptible to humans. We expect that a robust perceptual similarity metric should be tolerant to impercetible shifts so that its prediction is consistent with humans. Therefore, we tested on small shifts (no greater than 3 pixels) in our paper.
>
> We thank the reviewer for suggesting the resolution-dependent shift measurement. We have added that in our paper (in Section 4). For instance, a 3-pixel shift in our setting is equivalent to 1.2% as we used images of size 256 x 256 pixels.
>
> **Is the feature mapping of the proposed metric injective (or bijective)?**
>
> Re: We built our metric upon LPIPS by Zhang et al. and focused on investigating the various neural network elements involved in making the metric shift-tolerant. Since we closely follow the setup in LPIPS for our investigations, our feature mappings, like LPIPS, are not necessarily injective or bijective.

---

> ### Author Response · Authors · 2021-11-23
> **Comments on evaluations**
>
> Thank you for taking the time to review our paper. Below please find our responses.
>
> **The authors are encouraged to show the proposed metric is able to deliver SOTA quality prediction performance on standard IQA datasets.**
>
> Re: For an apples-to-apples comparison with DISTS, we train the LPIPS metric on the Kadid-10k dataset. We use the pre-trained model for DISTS that is trained on Kadid-10k and uses a VGG backbone network. The results on TID-2013 and LIVE datasets are as follows:
>
> *TID-2013 dataset* (below)
>
> | Metric           | SRCC | KRCC | PLCC |
> |------------------|------|------|------|
> | LPIPS(VGG)       | 0.86 | 0.66 | 0.87 |
> | LPIPS(VGG) ours  | 0.86 | 0.67 | 0.87 |
> | DISTS            | 0.83 | 0.64 | 0.85 |
>
> *Live dataset* (below)
>
> | Metric           | SRCC | KRCC | PLCC |
> |------------------|------|------|------|
> | LPIPS(VGG)       | 0.94 | 0.79 | 0.93 |
> | LPIPS(VGG) ours  | 0.95 | 0.80 | 0.94 |
> | DISTS            | 0.95 | 0.81 | 0.91 |
>
> Our shift-tolerant LPIPS metric improves the baseline LPIPS metric and outperforms DISTS on both datasets.
>
> **A more reasonable experimental setup is assigning shifted images the same MOSs and computing SRCC.**
>
> Re: We conducted the suggested study using the SRCC metric on the shifted version of the LIVE dataset. As reported in the table below, our upgraded LPIPS metric significantly improves the robustness against small shifts over the baseline LPIPS metric. Our results are comparable to DISTS.
>
> | Metric           | No shift | 1-pixel-shift | 2-pixel-shift |
> |------------------|:--------:|:-------------:|:-------------:|
> | LPIPS(VGG)       |   0.94   |      0.90     |      0.91     |
> | LPIPS(VGG) ours  |   0.95   |      0.95     |      0.94     |
> | DISTS            |   0.95   |      0.95     |      0.95     |
>
>
> Furthermore, we test the consistency of perceptual similarity metrics with the sensitivity of human perception to pixel shifts using the human responses we collected in our study. Our results (below) indicate that our upgraded metrics are more accurate than DISTS.
>
> |          **Metric**             |  **JND mAP%** |
> |:-------------------------------|:---------:|
> | SSIM                           |   0.722      |
> | LPIPS-Alex (Zhang et al., 2018) |   0.757     |
> | LPIPS-Alex$^*$                  |   0.740     |
> | LPIPS-Alex **ours**$^*$        |   0.771      |
> | LPIPS-VGG (Zhang et al., 2018) |   0.770     |
> | LPIPS-VGG$^*$                  |   0.769      |
> | LPIPS-VGG **ours**$^*$         |   **0.775**  |
> | DISTS                          |   0.766      |
> | PIM-1                          |   0.773      |
> $*$ Trained on BAPPS image patches of size 64 using author’s setup.
>
>
> **How is the performance of the proposed method to other mild geometric transformations, such as rotation and dilation?**
>
> Re: For this test, we use our shift-tolerant LPIPS(VGG) based metric trained on Kadid-10K. We rotate the distorted image by 1 deg clockwise. The metric’s SRCC decreases from 0.95 to 0.91, which is significantly better than the vanilla network, where the SRCC drops from 0.94 to 0.83. In addition, our shift tolerant LPIPS(VGG) metric is robust to dilation (by a factor of 1.5) as its SRCC does not decrease.

---

### Author Response · Authors · 2021-11-23
**To all reviewers**

We thank the reviewers for their thorough and constructive feedback. Considering the suggestions made by all the reviewers, we have updated our submission as follows:
1. Updated the related work section with a discussion on complex wavelet structural similarity index (CW-SSIM) and added its result in Tables 1 and 2.
2. Updated the related work section to better discuss DISTS.
3. Added more analysis regarding user responses and updated Appendix A.1 with more details regarding our user study.
4. Performed a JND test using the psychometric data from our user study to compare perceptual similarity metrics with the sensitivity of human perception to pixel shifts in Appendix A.1.1.
5. Evaluated our metric on other IQA datasets such as LIVE and TID-2013 and provide more comparisons with DISTS in Appendix A.4.
6. We modified the whole paper to incorporate comments from the reviewers while meeting the page limit.

---

### Decision · Program_Chairs · 2022-01-20

**Decision:**

Reject

**Comment:**

This paper introduces a perceptual similarity on top on the commonly used perceptual loss in the literature (LPIPS). The authors draw experiments highlighting that human perceptual similarity is invariant to small shifts, whereas standard metrics are not. The paper studies several factors (anti-aliasing, pooling, striding, padding, skip connection) in order to propose a measure on top of LPIPS achieving shift-invariance.

This paper initially received mixed reviews. RLHuY was positive about the submission, pointing out the relevance of the real human data and the studied factors for measuring the impact on shift invariance. RGQvy was slightly positive, but also raised several concerns on justification of the claimed properties, human perception experiments, and positioning with respect to data augmentation (PIM). RLHuY, an expert on the field, recommended clear rejection, pointing out missing references (including DISTS), the limited scope of the paper (shift invariance and tiny shifts). After rebuttal, RLHuY and RLHuY stuck to their positions ; RGQvy were inclined to borderline reject  because of unconvincing answers on comparison to data augmentation techniques.

The AC's own readings confirmed the concerns raised by RGQvy and RLHuY, and points the following limitations:

- The submission includes limited contribution and expected results: the studied modifications on neural networks' architecture, although meaningful, directly follow ideas borrowed from the literature. They are not supported by stronger theoretical analysis, and several insights related to accuracy or robustness remain unclear.
- Experimental results are contrasted, e.g. compared to data-augmentation: although these approaches are more demanding at train time, they do not induce any overhead at test time - in contrast to the proposed approach.

Therefore, the AC recommends rejection.